# Responses to Ecopollutants and Pathogenization Risks of Saprotrophic *Rhodococcus* Species

**DOI:** 10.3390/pathogens10080974

**Published:** 2021-08-02

**Authors:** Irina B. Ivshina, Maria S. Kuyukina, Anastasiia V. Krivoruchko, Elena A. Tyumina

**Affiliations:** 1Perm Federal Research Center UB RAS, Institute of Ecology and Genetics of Microorganisms UB RAS, 13 Golev Str., 614081 Perm, Russia; kuyukina@iegm.ru (M.S.K.); nast@iegm.ru (A.V.K.); tyumina@psu.ru (E.A.T.); 2Department of Microbiology and Immunology, Perm State University, 15 Bukirev Str., 614990 Perm, Russia

**Keywords:** actinobacteria, *Rhodococcus*, pathogenicity factors, adhesion, autoaggregation, colonization, defense against phagocytosis, adaptive strategies

## Abstract

Under conditions of increasing environmental pollution, true saprophytes are capable of changing their survival strategies and demonstrating certain pathogenicity factors. Actinobacteria of the genus *Rhodococcus*, typical soil and aquatic biotope inhabitants, are characterized by high ecological plasticity and a wide range of oxidized organic substrates, including hydrocarbons and their derivatives. Their cell adaptations, such as the ability of adhering and colonizing surfaces, a complex life cycle, formation of resting cells and capsule-like structures, diauxotrophy, and a rigid cell wall, developed against the negative effects of anthropogenic pollutants are discussed and the risks of possible pathogenization of free-living saprotrophic *Rhodococcus* species are proposed. Due to universal adaptation features, *Rhodococcus* species are among the candidates, if further anthropogenic pressure increases, to move into the group of potentially pathogenic organisms with “unprofessional” parasitism, and to join an expanding list of infectious agents as facultative or occasional parasites.

## 1. Introduction

A stressful environmental situation drives the need for expanded research on the features of microorganisms in polluted environments, that is, the so-called extremotolerant or stress-tolerant microorganisms. As components of natural ecosystems and native inhabitants of soils and waters, they play the role of the primary system of response to adverse or potentially dangerous environmental changes and initiate adaptive responses in the earliest stages. Stress-tolerant bacteria must harness a large arsenal of adaptive features to survive in the permanent struggle for a sustainable existence facing competition for the substrate, predation by higher organisms, increased impacts of xenobiotic compounds constantly released into the environments, and adverse abiotic factors.

Among the stress-tolerant species, actinobacteria of the genus *Rhodococcus* (phylum *Actinobacteria*, class *Actinomycetia*, order *Corynebacteriales*, family *Nocardiaceae*) (available online at: https://lpsn.dsmz.de/class/actinomycetia (accessed on 20 July 2021)) stand out due to their greatest variety of degraded xenobiotics and their complete mineralization of chemical pollutants to simple substances using alternative carbon sources (cometabolism), often followed by useful product formation [1,2,3].

A wide range of actinomycetologists focus on the study of polyextremotolerant rhodococci predominant in anthropogenically disturbed biotopes primarily due to their feasible applications in ecobiotechnologies. Although the study of *Rhodococcus* was, until recently, largely academic, it has now become more applied. This is obviously because, due to their metabolic flexibility, rhodococci do have not many “competitors” in terms of decomposing organic xenobiotics to inorganic products or low-molecular-weight organic fragments that can participate in the natural carbon cycle.

Considerable material is available on the degradation of priority pollutants by rhodococci to aid the understanding of their metabolic pathways. The number of publications is growing on new chemical agents degradable by rhodococci (Figure 1); and *Rhodococcus* genome projects have been launched [4,5,6]. The ability of rhodococci to decompose recalcitrant xenobiotics and resist their toxic effects, in addition to *Rhodococcus* survival strategies under the combined actions of ecotoxicants and other exogenous harmful factors, are well documented [5,7,8,9,10,11,12,13,14,15,16,17,18,19,20].

The immanent property of rhodococci is to synthesize cell components due to gaseous (C_3_‒C_4_) and liquid *n*-alkanes. Because of the capability of *Rhodococcus* to engage in oxidative transformation and degradation of natural and anthropogenic hydrocarbons, they are the least dependent on the external environment, avoid substrate competition, and survive in hydrocarbon-polluted environments. Thus, rhodococci are key players in maintaining the environment′s sustainability, active participants in the biogeochemical cycles of the biosphere, and significant contributors to the natural restoration of oil-polluted ecosystems and a carbon-free atmosphere of the Earth. This is their most general planetary function [21,22,23,24].

Our long-term geographically-tailored study of *Rhodococcus* diversity in anthropogenically loaded soils and aquatic ecosystems resulted in the gathering of factual material, namely, identification of numerous pure non-pathogenic cultures and descriptions of their characteristics. It was experimentally confirmed that the characteristic of a constant number of *Rhodococcus* not influenced by sharp seasonal fluctuations is a specific feature of hydrocarbon-oxidizing bacteriocenoses in oil-producing areas. In soils heavily polluted (up to 10 wt.%) with petroleum products, the average levels of ecologically significant species *R. erythropolis, R. globerulus, R. opacus, R. rhodochrous*, and *R. ruber* are 10^4^–10^5^ cells/g of soil, thus proving their role in the natural biodegradation of petroleum hydrocarbons [25,26,27]. The isolated strains are characterized by emulsifying abilities and biodegradative activities, not only towards aliphatic and aromatic hydrocarbons, and petroleum products, but also towards other recalcitrant and toxic pollutants—heterocycles, oxygenated and halogenated compounds, nitroaromatics, and organochlorine pesticides. They tolerate high (100–250 mM or more) concentrations of toxic metals and metalloids, and organic solvents (from 20 to 80 vol.%). The strains are active in a wide range of temperatures (from 4–15 to 40 °C and above), acidity (pH from 2.0 to 9.0), and humidity (from 15 to 50%), and are able to grow at high (2–6%) salt concentrations [14,20,26,28,29,30]. These *Rhodococcus* strains are the core of the Regional Specialised Collection of Alkanotrophic Microorganisms (acronym IEGM, available online at: http://www.iegmcol.ru (accessed on 20 July 2021)). The collection has been recognized by the Center for Collective Use (CKP_480868) and included in the National Research Facilities Registry of the Russian Federation [31,32]. The compiled fund of non-pathogenic strains of rhodococci is a helpful resource for identifying novel bioproducers of valuable substances, and biodegraders and biotransformers of complex organic compounds. *Rhodococcus* strains with active oxygenase enzymes are promising candidates for biotechnological processes, such as biodegradation of toxic pollutants and bioremediation of contaminated areas.

In this regard, there is a pressing need for new knowledge on universal and specific features of *Rhodococcus*, particularly those with induced oxygenase enzyme complexes, and new facts about their interaction with xenobiotic compounds. These facts would provide a more fundamental understanding of the role of this actinobacterial group in the functioning of the biosphere, and in removing or decreasing toxic components under the conditions of a destabilized environment. Finally, these species create prerequisites and opportunities for developing advanced biotechnologies of neutralization or efficient reuse of industrial wastes. Rhodococci are relatively new objects of environmental and industrial biotechnologies. Their metabolic potential for biodegradation and inactivation of complex pollutants, in addition to their mechanisms of stress resistance, are far from being exhausted. However, one should be conscious that some members of this genus are pathogens, and their number is gradually expanding, which clearly limits the practical application of rhodococci [33,34,35,36,37,38].

The range of *Rhodococcus* habitats is vast and diverse: these range from nutrient-rich (human and animal organisms, and plants) to oligotrophic (ground water, snow, and air) environments. Their adaptive traits include protective capsule-like structures and a lipophilic cell wall; colony dissociation and cellular polymorphism; temporarily dormant cyst-like cells and a low level of endogenous respiration (ensuring survival, e.g., upon prolonged starvation); production of carotenoids and extracellular glycolipids; nitrogen fixation in the presence of hydrocarbons; oligo- and psychrotrophy; acido-, alkalo-, halo-, xero-, thermo-, and osmotolerance; cell adhesion; and colonization of surfaces. Such a diversity of adaptive traits allows *Rhodococcus* accommodation in soil and aquatic environments, and possibly drove the appearance of epidemic variants in soil (water) and pathogenization of free-living forms. In extreme conditions of the polluted environment, rhodococci, as true saprophytic Gram-positive bacteria, are able to change their survival strategy in a manner that begins to exhibit traits of antagonism and pathogenicity [22,35].

In this review, actinobacteria of the genus *Rhodococcus*, typical mycolic acid-containing nocardioform bacteria, are used as an example to discuss the issues of common bacterial defense mechanisms (namely, ecological plasticity, population heterogeneity, ubiquity, long-term persistence in unfavorable conditions, and a wide range of tolerance to abiotic factors) against negative effects of anthropogenic pollutants and possible risks of pathogenization of free-living bacterial forms. Special attention is paid to adaptive cell modifications of rhodococci exposed to hydrocarbon contaminants as putative “universal” pathogenic factors.

## 2. Ubiquity of *Rhodococcus* and Existence of Pathogenic Species

### 2.1. Free-Living Rhodococci

Extremotolerant (psychroactive and thermotolerant, acid- and alkalotolerant, and halo- and haloalkalotolerant) *Rhodococcus* species are known mainly as typical inhabitants of soil environments ranging from tropical to Arctic soils, including the remote areas of Kamchatka and Yakutia. Rhodococci are co-members of the microbiota of natural, transformed (agrocenoses), and urbanized (urbanocenoses) ecosystems. They have been isolated from extreme and unique environments, such as cold polar deserts, Antarctic and alpine soils, annually freezing and thawing tundra soil, salt marshes, and dry desert sand, in addition to mangrove sediments, snow, air, core, and a variety of anthropogenic habitats [12,25,28,30,32,39,40,41,42,43,44,45,46,47]. They inhabit different water bodies, such as lakes and ponds, groundwater, and mineral and stratal waters, and are found in the bottom sediments of northern seas contaminated with petroleum products, and in oil fields [48,49,50,51,52,53,54,55,56,57,58].

When dwelling in soil or water, bacteria are directly affected by a variety of abiotic factors subject to strong periodic (daily, seasonal, long-term) and non-periodic fluctuations. For example, diurnal temperature fluctuations in temperate soils can exceed 15 °C [59], with greater seasonal fluctuations. Given the huge fluctuations in environmental factors, it should be assumed that ubiquitous rhodococci have high ecological flexibility and tolerance.

However, knowledge about the distribution and seasonal dynamics of freshwater and marine populations of *Rhodococcus* is relatively scarce, and there are little data on their physiological and ecological roles in the communities. Whether rhodococci are their permanent inhabitants or transported from the soil has not been elucidated. There is only an assumption that rhodococci in the dormant coccoid stage pass into freshwater and marine environments [60].

### 2.2. Phytopatogenic Rhodococci

Rhodococci have been detected in rhizosphere and phyllosphere microbiota [61,62,63,64,65]. Associated with plants, rhodococci can not only stimulate their productivity [66], but also cause pathology. Phytopathogens *R. corynebacterioides* and *R. fascians*, which are causative agents of bacteriosis in fruit and berry plants, have been described. Initially, *R. fascians* was an epiphyte that turned into an endophyte [67,68]. *R. fascians* induces fasciation and leafy gall disease in a variety of plants (namely, 87 genera spanning 40 plant families), including the recently emerged Pistachio Bushy Top Syndrome (PBTS) [67,69,70,71,72,73,74]. It is known that persistence of *R. fascians* cells in the tissues of infected plants and manifestation of virulent properties are due to the linear plasmid pFiD188 present in cells [67,68,72,75]. The plasmid contains genes encoding proteins of the bacterial cell transition from the epiphytic to the endophytic state, and the subsequent synthesis of specific cytokinins that change the plant metabolism. It is still debated which loci in the plasmid leads to manifestation of virulence in *R. fascians*. Creason et al. [68] believed that horizontal acquisition of just four functions involved in secondary metabolism (*att*), gene transcription (*fasR*), cytokinin biosynthesis (*fasD*), and cytokinin activation (*fasF*) is sufficient for *Rhodococcus* to infect plants. It should be noted that the presence of the virulence plasmid is essential for the pathogenicity of *R. fascians*. Based on the analysis of genome sequences from over 80 plant-associated *Rhodococcus* isolates, it was found that horizontal gene transfer provides an evolutionary transition from their symbiotic to their pathogenic existence [76]. Endophytic cells of *R. fascians* multiply in the intercellular space of the galls, in addition to inside the cells. The persistence of *R. fascians* cells in the tissues of infected plants is associated with the expression of the glyoxylate cycle genes and glycine metabolism genes, allowing the bacteria to use plant metabolites as carbon sources. The pFiD188 plasmid was found not to affect colonization. *R. fascians* cells are surrounded by mucus, which apparently consists of extracellular polymeric substances. The mucus creates beneficial conditions for the cells’ adhesion and protects them against drying. The mechanisms of adhesion of bacteria of this species in the epiphytic form to the surface of plant leaves have not been studied.

### 2.3. Animal and Human Rhodococcus Pathogens

Typical soil rhodococci “*Rhodococcus equi* (*Rhodococcus hoagii*/*Prescottella equi*)” have long been isolated from the lung tissues of 3 month old foals and are now known as a facultative intracellular pathogen, not only in young domestic animals, but also as a serious opportunistic human pathogen. For many years, “*R. equi*” representatives were isolated from soils of all continents, except for Antarctica. They are the causative agents of slowly progressing pneumonia (“rhodococcal pneumonia”, “rhodococcal infection”, “foal rhodococcosis”) and subcutaneous abscesses most frequently observed in HIV-infected patients with a compromised immune system [77,78,79,80]. Several cases of nosocomial infections caused by “*R. equi*” (meningitis, pneumonia, endocarditis, and keratitis) have been described [33,81,82,83,84,85,86,87,88]. For animals and humans, the main route of exposure is inhalation of soil dust. Infection with the pathogen “*R. equi*” occurs through the lungs, digestive tract, or damaged skin. Rhodococcosis is a chronic and recurrent disease that is difficult to treat.

Based on the results of “*R. equi*” genome studies, hallmarks that distinguish this zoonotic pathogen from non-pathogenic rhodococci have been identified. “*R. equi*” is genetically homogeneous, and has a small (about 5.0 Mb) genome, which lacks the extensive catabolic and secondary metabolic (e.g., phosphoenolpyruvate:carbohydrate transport system) complement of environmental rhodococci [35,37]. The pathogenicity of “*R. equi”* is mediated by Virulence Associated Proteins (VAPs) encoded on the acquired pathogenicity islands (PAIs) of host-adapted conjugative virulence plasmids [89]. “*R. equi*” isolates from equine, porcine, and bovine samples carry different types of plasmids called pVAPA, pVAPB, and pVAPN, respectively [90].

Within equine pVAPA plasmid PAI, the main virulence factors are the *vapA, -C, -D, -E, -F*, -*G*, and *-H* genes. Of these seven *vap* genes, *vapA* has been shown to be a key virulence factor encoding a surface-localized VapA protein that is important for bacterial replication in host macrophages. *“R. equi”* cells enter the host through the respiratory system and infect alveolar macrophages. The binding of “*R. equi*” to macrophages is mediated by both specific and nonspecific factors: the complement system, Mac-1 receptor, hemagglutinin, and hydrophobic interactions. In response to infection, the activated macrophages produce large amounts of reactive oxygen species, such as H_2_O_2_ and superoxide anions. To resist oxidative stress, members of an intracellular “*R. equi*” pathogen possess specific mechanisms. Expression of their plasmid genes is triggered by changes in pH, decreased iron availability, increased oxidative stress, and increased temperatures. The major virulence-determinant VapA protein expression increases with oxidative stress and a low pH value. VapA shifts the pH values to neutrality inside the macrophage and promotes the exclusion of the vacuolar-type ATPase from the phagosome. Further VapA can be transported from phagosomes to lysosomes, making their membranes more permeable to protons. In addition, “*R. equi*” has genes encoding heat shock proteins, Clp proteases, catalase, superoxide dismutase, alkyl hydroperoxide reductase, and mycoredoxins [35,77,91,92,93]. The pVAPB plasmid consists of *vapB, -J, -K1, -K2, -L*, and *-M*, whereas pVAPN carries *vapN, -O, -P*, and *-Q* genes. *VapK1, vapK2, vapB*, and *vapN* are functionally equivalent to *vapA* [90,94,95].

Another important mechanism that determines the virulence of rhodococci is adhesion. Letek et al. [35] carried out a detailed analysis of the “*R. equi*” 103S genome and identified the putative adhesion molecules of this species: (1) ibronectin-binding proteins REQ01990, 02000, 08890, and 20840—homologues of the protein Fbp/antigen 85 of *M. tuberculosis*; (2) type IVb cytoadhesive pili of the Flp-pili subfamily of Gram-negative bacteria for attachment to epithelial cells and macrophages. They apparently were introduced into the “*R. equi*” 103S genome through horizontal gene transfer in the form of an *rpl*-fragment of nine genes encoding the biogenesis of these structures; (3) adhesins REQ38170 and REQ31340—homologues of mycobacterial cytoadhesins, i.e., heparin sulfate-binding hemagglutinin HbhA and multifunctional histone-like/laminin- and glucosaminoglycan-binding protein Lbp/Hlp; (4) adhesin REQ34990, a protein with unknown function with the FAS1/BigH3 domain responsible for cell adhesion using integrins. Regarding cytoadhesive pili, it should be noted that pili have not been described in rhodococci. Letek et al. [35] reported that the presence of pili and their involvement in “*R. equi*” cell adhesion was experimentally confirmed; however, the description of the experiments was not published in the available scientific literature.

The phylogenetic analyses have shown that “*R. equi*” has a high degree of core genome (conserved gene set) similarity with non-pathogenic environmental rhodococcal species [96]. There are genes involved in protein biosynthesis, metabolism, transport, and signal transduction, which can be associated with both adaptation and virulence. Letek et al. [35] also noted that virulence-related genes in “*R. equi”* are conserved in environmental rhodococci or have homologs in non-pathogenic actinomycetes. It was speculated that “*R. equi*” evolved due to the co-option of core actinobacterial functions (internal virulence resources) originally selected in a non-host environment through horizontal gene transfer. This leads to the transition of a commensal organism into a pathogen due to the co-option of pre-existing virulence-associated core genes. Ecologically significant rhodococci have a wide range of stress-managed strategies, which, under certain conditions, can serve as a basis for transition to a pathogenic lifestyle [76]. Moreover, the spread of antibiotic resistance in the environment poses a particular risk. “*R. equi*” is a reservoir for the antibiotic resistance genes and is involved in the dissemination of resistance genes in nature through highly transferrable conjugative plasmids, which leads to the emergence of multidrug resistance among non-pathogenic rhodococci [89,97].

Relatively recently, the first descriptions were made of human diseases caused by other representatives of rhodococci, i.e., non-*equi Rhodococcus* (*R. erythropolis, R. globerulus, R. gordoniae*, and *R. rhodochrous*) [33,34,36,98]. This was largely due to the technical difficulties faced by ordinary clinical microbiological laboratories when diagnosing these pathogens. Only with modern diagnostic methods (including 16S sequencing, etc.) has their correct identification become possible [98].

Due to the prospects of exploiting rhodococci in open systems (soil, water, and sewage treatment plants), combined with the large number of new species of *Rhodococcus* spp. described in the last decade, an urgent task has arisen to determine the potential risks associated with their introduction into natural environments.

## 3. Adaptive Cell Modifications of Rhodococci Exposed to Hydrocarbons and Other Environmental Pollutants

A recent review [20] discussing various stress effects on *Rhodococcus* populations summarizes physiological and phenotypic reactions of bacterial cells in response to environmental stresses, in addition to the responses to the effects of toxic metals, antibiotics, and other organic compounds. Previously, separate studies also considered effects of stress reactions of rhodococci on their persistent potential [7,16,22].

Although the complex of protective mechanisms is tangled and diverse, there is a certain systematic nature underlying the organization and operation of this system. Morphological and physiological mechanisms of adaptation are interrelated with molecular (biochemical) mechanisms of protection, including mechanisms of DNA repair. All of these interactions are aimed at maintaining and increasing the resistance of *Rhodococcus* to adverse environmental factors. For example, when grown on hydrocarbons and their derivatives (cyclohexane, naphthalene, diesel fuel, etc.), *R. erythropolis* exhibits an increased resistance to the concomitant strong oxidative stress arising from the toxic substrate decomposition and leading to the accumulation of highly reactive oxygen species, H_2_O_2_, and highly toxic lipid peroxides. Additionally, the induced activity of antioxidant enzymes due to the overexpression of antioxidant-encoding genes (cytochrome P450, catalase, superoxide dismutase, alkyl hydroperoxide reductase, and the protective protein RecA) involved in DNA repair, detoxification of reactive oxygen species, and, finally, in detoxification of pollutants, has been observed [99,100]. Morphological changes occur, particularly changes in size and shape, in addition to an increase in the number of cytoplasmic polyphosphate inclusions (volutin), hypertrophy of individual organelles, and the formation of multicellular aggregates, biofilms, and other spatial structures. Under oxidative stress, rhodococci accumulate carotenoids as additional antioxidants that utilize singlet oxygen [101]. During cell aggregation, the expression of the global regulator gene, a sigma subunit of RNA polymerase SigF3, increases. The global regulator gene controls the expression of genes that enable bacterial cell adaption to changing environmental conditions [102].

Therefore, when influenced by adverse environmental factors, rhodococci flexibly and effectively self-regulate the cell functioning due to the cross-combination of adaptation mechanisms that ensure the maintained cell integrity and high stress tolerance. All *Rhodococcus* responses to recalcitrant hydrophobic compounds are of a multifaceted adaptive nature, manifested at different levels of the cell organization, and are essentially similar, general, and universal.

The most frequently detected disorders occurring in the early stage of non-specific reactions of cells to destructive effects include: (1) changes in the affinity of the cell surface to hydrophobic damaging agents and in surface properties, including the cell wall hydrophobicity; (2) morphometric modifications, including the average size of vegetative cells, and the relative area and relief of the cell surface; and (3) changes in the integral physical and chemical parameters of cells, particularly electrokinetic characteristics [22,103,104,105,106,107,108,109,110].

Rhodococci adapt to hydrophobic pollutants generally by forming separate multicellular aggregates (Figure 2). For example, when rhodococci were grown in the presence of liquid *n*-alkanes, planktonic microsized (25–70 µm in diameter) cell aggregates (microcolonies) of an elongated or rounded shape consisting of viable cells were formed during the first day (Figure 2A,B). On days 3–4, macroscopic compact multicellular (biofilm) formations (Figure 2C), or mucosal floccular strands (flocks) up to 1 cm in size diffusely located throughout the entire volume of the medium were observed. The spontaneous formation of biofilm clusters (cell aggregates, microcolonies) in the form of dense granules (up to 5 mm in diameter) seen by the naked eye, floating freely in the medium, and remaining until the end of the experiment, reduced the number of suspended cells in the culture broth (Figure 2D). By further self-immobilization, larger mushroom-like aggregates with a uniform structure passively floating in the liquid nutrient medium were formed (Figure 2E,F). In these cell clusters, the spatial arrangement of cells usually follows the rule of minimum diffusion distance, which ensures the fastest mass transfer between cells. Adhesion forces act at a distance of <20 nm [111].

Aggregation is an effective phenotypic adaptation that allows rhodococci to grow successfully in suboptimal environments. This is one of the factors related to bacterial persistence and pathogenicity. Such a “cooperative cell system” provides the coordinated functioning of numerous associated cells and allows the population to adapt and grow in “harsh” conditions, in which individual cells are not able to reproduce and biodegrade pollutants.

The cell aggregation and the ability to form bulky “oversized” morphotypes, including the presence in the life cycle of “inedible” large branching threadlike cells (greater than the size of a prey for a potential predator), furnish the *Rhodococcus* population with the protection from a range of predators and parasites, such as microbivorous free-living protists (amoebae, flagellates, and ciliates) and bacteriophages [112]. Issues concerning the experimental study of the ecological interactions between bacteria and protozoa, and specific strategies of rhodococci to mitigate the risk of predation by protozoa in soil, freshwater, and marine ecosystems, are still subject to detailed consideration [113,114]. Future research should focus on the complex role of protists and the reduced vulnerability of rhodococci to ecologically relevant species in planktonic microbial communities.

Pronounced heteromorphism, morphogenetic alterations of vegetative cells, a complex development cycle of *Rhodococcus*, and finally, the rigid cell wall structure, are clearly relevant to adaptation. The advantages of these attributes for survival are obvious and can be considered as mechanisms of successful adaptation that determine the stability of *Rhodococcus* populations under variable environmental conditions.

Exposed to ecotoxicants, rhodococci can vary the fatty acid composition of membrane lipids, including mycolic acids and phospholipids, to maintain the level of viscosity of their membranes in the liquid-crystalline state [108,115]. Lipids are one of the few groups of complex organic substances for which the composition can be adequately regulated by the bacterial cell in response to suboptimal growth conditions by changing the structure and relative amount of membrane fatty acids [116]. The key role in determining the fluidity and permeability of *Rhodococcus* cell membranes is due to the composition of fatty acids, which can vary in the length and degree of saturation of the hydrocarbon chains depending on the organic substrate consumed [29,117,118,119].

*Rhodococcus* are capable of shifting to specific resting (viable but nonculturable, VBNC) forms [11,120] and accumulating large amounts of storage substances, for example, triacylglycerols, esters wax, and polyhydroxyalkanoate, as reservoirs of carbon and chemical energy [52,103,121]. The accumulated neutral lipids are not only a basis for the biosynthesis of mycolic acids and regulation of membrane fluidity, but also a source of endogenous water [7,30], which allows rhodococci to tolerate soil drought and high temperatures [8,12,52,121].

Proton- and sodium-dependent efflux pumps, which are energy-dependent systems for removing toxic compounds from the cell, are involved in the resistance of *Rhodococcus* to the negative effects of pollutants [108]. There are efflux pumps that are active against a single substance or a class of substances, and so-called Multidrug Resistance (MDR) efflux pumps for antibiotics of different classes. The functioning of efflux pumps (e.g., MDR) may also be crucial for the survival of rhodococci in natural coexistence with antibiotic-producing microorganisms (e.g., *Streptomyces* spp.) [22]. When grown on hydrocarbons, rhodococci are characterized by an increased resistance to a wide range of antibiotics (aminoglycosides, lincosamides, macrolides, etc.). This effect is probably due to a decrease in the ionic permeability of the outer cell membranes for hydrophilic antibiotic molecules [122,123]. Dissemination of MDR is of special concern, because the pRErm46 plasmid that carries MDR-encoding genes is easily transferred from a zoonotic pathogen “*R. equi”* to other actinobacterial species in the environment, and, eventually, to pathogens of humans or other animals [89]. In addition, high competitiveness of rhodococci is provided by quorum-quenching activity against plant and human Gram-negative pathogens by degrading a wide range of signal substrates (lactones, quinolones, etc.) [124,125].

The most vulnerable target for the damaging effects of ecotoxicants and other adverse environmental factors is the peptidoglycan. All adaptation processes in *Rhodococcus* aimed at protecting or isolating the peptidoglycan structure should be considered as mechanisms of bacterial persistence. To survive in unfriendly environmental conditions, the bacterial cell seeks to protect its peptidoglycan in several ways.

(1) Mechanical protection by a mucosal microcapsule on the cell surface less than 0.2 µm thick to screen the exposed peptidoglycan (Figure 3C,D). The formation of these surface structures in the form of a loose layer of microfibrils of polysaccharide nature, which are stained with ruthenium red, is an adaptive feature. The polysaccharide matrix enveloping the cells also plays an important role in the adhesion of rhodococci on the substrate, promotes cell aggregation and biofilm formation, and additionally protects the cells from adverse external exposures, including phagocytosis [126,127,128,129]. Protection against phagocytosis is a prerequisite for saprophytic bacteria to exist in aquatic and soil communities. *Rhodococcus* resistance to phagocytosis due to peptidoglycan shielding by capsule formation, similar to that of pathogens, is not only an adaptation to survive in environmental communities, but also the pathogenic potency.

Our earlier studies showed high antibiotic resistance of rhodococci grown in the presence of gaseous hydrocarbons and located in biofilms, resulting from poor penetration of most antibiotics into biofilms [122,130]. According to the U.S. Centers for Disease Control and Prevention (CDC), up to 65% of bacterial infections occur with the formation of biofilms [131].

(2) Protection of the peptidoglycan by specific accessory structures—rounded or elongated pineal appendages (up to 40 nm in diameter and up to 600 nm long) attached to and located irregularly on the outer surface of the cell wall (Figure 3A). The purpose of multiple pineal appendages is unclear. It can be assumed that they are multifunctional and most likely provide a cytoadhesive function, i.e., interaction between cells, retention of cells in microcolonies and films, and cell attachment to (a)biotic substrates, or possibly their protection from phagotrophic protists, in addition to intercellular communication. These surface organelles appear to not only have adhesive properties and perform aggregate functions, but can also be associated with DNA transport by horizontal gene transfer and provide a possible exchange of cytoplasmic material.

(3) Modification of cell membrane lipid composition and enhancement of hydrophobic properties and interactions (Figure 3B), in addition to the production of surfactants that solubilize hydrophobic pollutants [103,104,117,132,133,134]. The study of rhodococci by cryofractography showed the presence of a large number of rounded structures on the cell chips surrounded by a roller and corresponding to fat droplets (see Figure 3B). Representatives of *R. rhodochrous* and *R. ruber* are the richest in fatty cellular inclusions. The identified inclusions in *Rhodococcus* spp. cells apparently serve as reserve substances providing a selective advantage to rhodococci under limited growth conditions in natural environments. The total lipid content and composition in the cell wall play a key role in the sorption of liquid hydrocarbons [135,136]. For example, in *R. ruber* (IEGM 324, IEGM 333, and IEGM 371 strains) grown in a mineral salt medium supplemented with *n*-hexadecane, approximately a two-fold increase in the number of total lipids (17–22% of the dry cell weight) was found compared with that (8–14%) of cells grown in the nutrient broth [115]. Furthermore, an increase in straight chain saturated fatty acids and neutral phospholipids (particularly cardiolipin and phosphatidylethanolamine) was observed, which apparently leads to an increase in the degree of cellular adhesion to nonpolar hydrocarbons [111,115,122,137].

The above complementary strategies allow *Rhodococcus* to efficiently respond to stresses caused by environmental pollutants, and to preserve the integrity of cells, their metabolic activity, and their interaction with the environment.

### 3.1. Adhesion, Cellular Autoaggregation, and Colonization as Survival Strategies

An important role in rhodococci resistance to toxic xenobiotics and other unfavorable conditions is played by autoaggregation (cohesion), i.e., grouping of separate cells into large clusters (into a single multicellular organism) upon chemotaxis-mediated contacts. One main factor of aggregation is adhesion (direct contact of a bacterial cell with a substrate required for a bacterial population to develop and colonize surfaces) and specific adhesins. These are individual surface adhesive proteins that are part of the cell appendages and induce active migration of cells to the center of aggregates, and contribute to a significant reduction in the diffusion distance between the substrate and the cell.

Adhesion and autoaggregation are typical responses of saprophytic rhodococci to chemical agents, which are not favorable for bacterial growth [10,138]. Upon aggregation, the complexity of interactions within the cooperating cell system increases dramatically compared to that of planktonic forms, and therefore a specific microenvironment is formed that provides efficient functioning of the entire cell population. Such aggregations of hydrophobic cells apparently contribute to the accelerated cooperative attack of oxidative enzymes on the ecotoxicant.

As shown above, in the presence of hydrophobic compounds (hydrocarbons, in particular), rhodococci form isolated multicellular aggregates of different sizes and irregular shapes (see Figure 2). The main role of aggregation is primarily to protect bacterial cells from the toxic effects of chemical compounds. This is confirmed by the fact that aggregation is usually more pronounced in *Rhodococcus* spp. strains that are less resistant to pollutants and is enhanced upon elevated concentration of the pollutant in the medium. Thus, in the presence of dehydroabietic acid (abieta-8,11,13-triene-18 acid, C_20_H_28_O_2_, i.e_.,_ a resin acid and an ecotoxicant, and one of the dominant components in pulp and paper wastewater), a non-resistant *R. erythropolis* IEGM 267 forms large (up to 5 mm in diameter) cell aggregates, whereas the resistant *R. rhodochrous* IEGM 107 forms small (up to 0.5 mm in diameter) cell aggregates [110]. In a medium with diclofenac (2-(2-[2′,6′-dichlorophenyl]-amino)-phenylacetic acid in the form of a sodium salt, C_14_H_10_Cl_2_NNaO_2_)—a widely available non-steroidal anti-inflammatory drug frequently used in human and veterinary medicine, and a pharma pollutant—the maximum aggregation of *R. ruber* IEGM 346 cells was observed at a high (50 mg/L) diclofenac concentration [5] (Figure 4D and Figure 5C). Under the same conditions, the maximum distortion of the cell morphology was recorded: a change in the cell shape; enlargement of the average size; and decreases in the cell surface area and the surface area-to-volume ratio (Figure 4B,C). This is the defense mechanism that enables the bacteria to reduce the amount of the toxicant coming into the cells. Figure 4C demonstrates that at a high diclofenac concentration, there were numerous dead cells (fluorescent in red) indicating the toxic effect of this compound. Exposed to diclofenac, single cells suffered the damaged integrity of their peptidoglycan layer (Figure 5B), accompanied by the release of cytoplasm into the medium, and, as a result, the appearance and accumulation of dead cells in the sample. Nevertheless, the process of diclofenac bioconversion did not stop [5].

This observation may be explained by the phenomenon of an “altruistic” (programmed) cell death. Microorganisms are biosocial organisms, and many of them can transit from a single- to multicellular organizations (microbial colonies, biofilms, and aggregates). A multicellular structure of a bacterial population provides significant advantages: reliable protection against unfavorable environmental factors, increased genetic diversity, and the greatest availability of food sources. A bacterial population acting as a multicellular organism exploits the programmed cell death by sacrificing part of the colony to support the survival of the remaining cells [139]. In the presence of the recalcitrant chlorine-containing polycyclic compound diclofenac, the death of a part of the population and subsequent release of intracellular components not only supplies the remaining rhodococci with food sources and saves energy, but also contributes to implementing the successful adaptation strategies and, ultimately, to detoxifying and biodegrading this pharma pollutant. A similar observation was made in the presence of other complex organic compounds, e.g., betulin (lup-20(29)-ene-3,28-diol, C_30_H_50_O_2_, a pentacyclic triterpenoid of the lupan type present in birch bark and used in the synthesis of anti-inflammatory, hepatoprotective, antitumor, antiviral, antimalarial, and antibacterial drugs) [13,140], and dehydroabietic acid, an ecotoxic pollutant [110].

Cell aggregation is the result of changes observed at the membrane and cell wall levels [29]. Therefore, for a deeper insight into bacterial mechanisms of adaptation to toxicants, the morphometric parameters of cells, their nanogeometric characteristics, and an electric charge are essential.

### 3.2. Changes in the Morphometric Parameters of Cells

The shape and size of rhodococci play a crucial role in their interaction with ecotoxicants, for which a determining factor is the surface of their cell wall.

A detailed study of rhodococci interaction with different organic compounds was performed using a combined scanning system consisting of an Asylum MFP-3D-BIO atomic force microscope (AFM, Asylum Research, Goleta, CA, USA) and an Olympus FV1000 confocal laser scanning microscope (CLSM, Olympus Corporation, Tokyo, Japan). Using a combined AFM-CLSM scanning system, the responses of living cells to ecotoxicants, including rearrangements of the cell surface structures, were studied in a real-time mode and with a high spatial resolution that an optical microscopy cannot provide.

A characteristic “response” of alkanotrophic rhodococci to toxic substances is a shift in the morphometric parameters (namely, length and width, area, and roughness of the cell surface). The destructive effects of the above-mentioned diclofenac and dehydroabietic acid on rhodococci were manifested in a significant (*p* < 0.001) decrease in the relative surface area (area-to-volume ratio, S/V) of bacterial cells (Table 1). A decline in the S/V value is probably important for bacterial resistance to a toxic pollutant by reducing the cell surface contact with an ecostressor. By increasing their size, bacteria reduce the relative area of cell surfaces known to be the main target for aromatic compounds acting as membrane-active toxins [141,142]. Conversely, in the presence of less toxic substrates (e.g., betulin and low concentrations of diclofenac), the S/V values were opposite: *Rhodococcus* contact area with the substrate increased for its better absorption and assimilation [143]. Additionally, effective biotransformation and biodegradation of toxicants were accompanied by a decrease in the average cell size. This is a paradox of nature: *the smaller, the more productive* (*or stable*). The growth and reproduction energy of living organisms and the biomass formed are inversely proportional to their size.

Here, and in Table 2, cells were grown for 7–10 days [5,109,110].

Changes in bacterial size and surface area occur due to changes in the cell shape. At high diclofenac concentrations, for example, cells are heteromorphic, with a disturbed division process or with a defective cell wall, and are enlarged mainly due to swelling (see Figure 4C and Figure 5B). At low diclofenac concentrations, the number of cells with structural irregularities and heterogeneous morphological features increases: cells are decreased in size, become more oval, are often highly swollen, cigar-, flask-, or bean-shaped, and have convex edges (see Figure 4B and Figure 5C). It can be assumed that such changes in rhodococci are associated with the disturbed cell wall synthesis or cell division. Many cells affected by pharmaceutical pollutants and then placed in friendly conditions can repair their damaged structures and control mechanisms.

Studying the nanogeometry of living cells has revealed increases in the root-mean-square roughness (an integral dispersion parameter of surface irregularities) and in the amplitude with all of the substances tested (see Table 1). A change in the roughness of the bacterial cell wall indicates global changes occurring in cells under adverse environmental conditions. First, the roughness increases the S/V and, accordingly, the adsorption of the substrate by cells [144]. In turn, surface complexation with toxicant ions changes the cell architecture, leading to an increase in the cell wall roughness [145]. Second, an increase in roughness may indicate possible rearrangements in the biosynthesis of peptidoglycan, accompanied by a slowdown in cell division and cell lysis [146]. Third, the increased cell roughness is a result of extracellular polymer secretion and changes in the lipid composition of the cell wall, which increases the Van der Waals forces, and thus promoting better adhesion of cells to each other and to substrates [146]. The study of viable but nonculturable *R. biphenylivorans* T9 cells in the presence of the antibiotic norfloxacin revealed an increase in the cell wall roughness proportional to the increase in biofilm formation [147].

### 3.3. Change in the Zeta Potential of Cells

Interaction with ecostressors results in changing the physicochemical characteristics of *Rhodococcus* cells, including the electrokinetic potential (zeta potential), which quantitatively reflects the electrosurface properties of a cell [148]. The zeta potential plays an important role in maintaining cellular functions, and its change due to electrostatic interactions between the cell and environmental factors can lead to a modification of the permeability of the cell membranes and, potentially, cell death. Table 2 presents the results of measuring the zeta potential of the cell surface of rhodococci exposed to the above-described substrates.

In their native state, rhodococci have negative zeta potential values due to the presence of negatively charged lipid molecules, lipoglycans, in the bacterial cell wall. Rhodococci have an unusual structure and composition of the cell wall compared to other Gram-positive bacteria. It is dominated by complex specific lipids, which include 2-alkyl-3-hydroxy branched fatty acids responsible for the formation of an external impermeable lipid barrier [149,150]. Cultivation of rhodococci in the presence of terpenes (betulin and dehydroabietic acid) leads to an increase in negative values of the zeta potential. It is known that higher absolute values of zeta potential determine the colloidal stability of the cell population [151]. Apparently, an increase in the negative values of the zeta potential indicates the presence of a cell-bound trehalolipid biosurfactant. This biosurfactant has an anionic nature due to the carboxyl groups present and, therefore, can increase the negative charge of the bacterial cell surface. An increase in the zeta potential, and thus an increase in the stability of the bacterial cell system, provides a high rate of mass exchange between the cells, the medium, and the degraded compound [151].

A different picture was observed in the presence of pharmaceutical pollutants. Under diclofenac exposure, an increase in zeta potential was noted apparently due to the interaction of cations in the pharmaceutical molecules with the carboxyl groups of mycolic acids in the rhodococcal cell wall.

It is known that shifting the zeta potential towards neutrality may lead to destabilization of cell membranes and cell lysis [148,152]. During this period, a significant number of damaged and lysing cells was noted. According to some data, antibiotics contribute to changes in the electrokinetic potential of the bacterial cell surface, further accompanied by the violated cell division (by targeting the Min proteins) and morphology [153]. Changes in the morphology of rhodococci in the presence of pharmaceutical pollutants were confirmed by AFM scanning. Interestingly, partial cell death did not stop diclofenac biodegradation [5]. According to some data, a decrease in the electrokinetic potential may indicate an increase in toxicants′ adhesion to cells, and thus contribute to their degradation [154]. An improved adhesion was accompanied by an increase in cell wall hydrophobicity. The Salt Aggregation Test revealed that, in the presence of pharmaceuticals (diclofenac and a spasmolytic drotaverine), hydrophobic interactions between rhodococci were enhanced, promoting the development of cellular aggregates. Additionally, rhodococci produced stable microaggregates under a low (0.6 M) concentration of ammonium sulfate, indicating the high hydrophobicity of their cell surface [5].

## 4. Conclusions

*Until recently, the image of an “enemy” (less often, a “companion”) dominated in the human–microbe relationships, now it becomes obvious that it is necessary to establish “peaceful coexistence” with this huge world*.[155]

The goal of this review is to highlight the structural, physiological, and biochemical features of a metabolically versatile group of nocardioform bacteria—representatives of the genus *Rhodococcus*, which are well-known biodegraders of hydrocarbons and other lipophilic organic compounds. These bacteria occupy one of the dominant positions in anthropogenically disturbed biotopes, and participate in their attenuation and restoration. The relative simplicity of the *Rhodococcus* cell structures is in harmony with the amazing perfection of their biological organization, and with their ability to form peculiar cellular adaptations having deep impacts primarily on cytology and physiology of rhodococci.

These materials substantiate the idea that the adaptive reactions of rhodococci to the negative effects of ecotoxicants are complex. In the presence of ecotoxicants, the developing *Rhodococcus* population changes smoothly towards the formation of more stable forms adapted to new stressful conditions. In this state, *Rhodococcus* cells are characterized by significant universal morphological changes. The discovered natural means of maintaining constant intracellular conditions in rhodococci exposed to various pollutants are considered to be mechanisms of their adaptation to the changing environment. They provide an improved tolerance of rhodococci to xenobiotics and the ability to change their survival strategy, and show signs of pathogenicity. The high competitiveness and survival of *Rhodococcus* species in any environment, in addition to their pathogenic potential, are determined by the ability of rhodococci to adhere to, aggregate, and colonize surfaces, and to change their lifestyle (from unicellular to multicellular, and from saprotrophic biodegraders to plant pathogens and animal intracellular parasites); the long-term persistence in unfavorable conditions; diauxotrophy; the formation of cyst-like cells and capsule-like structures; the enhanced cell surface hydrophobicity; ubiquity; and the antagonistic activities of some rhodococci. In medicine, the adaptive mechanisms are commonly referred to as “pathogenicity factors”, with adhesins and adhesion being particularly important as the mechanisms that trigger an infectious process.

Amid the pathogenization of saprotrophs under conditions of man-made pollution, the range of potentially dangerous microorganisms with “unprofessional” parasitism is expanding. There is a certain blurring of boundaries between pathogens and non-pathogens, and the idea of “universal” pathogenic factors of microorganisms is becoming more established. Moreover, it becomes increasingly difficult to predict which group of today′s saprophytes will join the list of pathogenic agents of disease tomorrow. At present, there is only an accumulation of research data on *Rhodococcus* survival strategies. The mechanisms of possible pathogenization of saprotrophic rhodococci have not been studied at the level of genome functioning and regulation of their metabolism. This has yet to be undertaken using modern genomic and post-genomic technologies with the use of system analysis.

## Figures and Tables

**Figure 1 pathogens-10-00974-f001:**
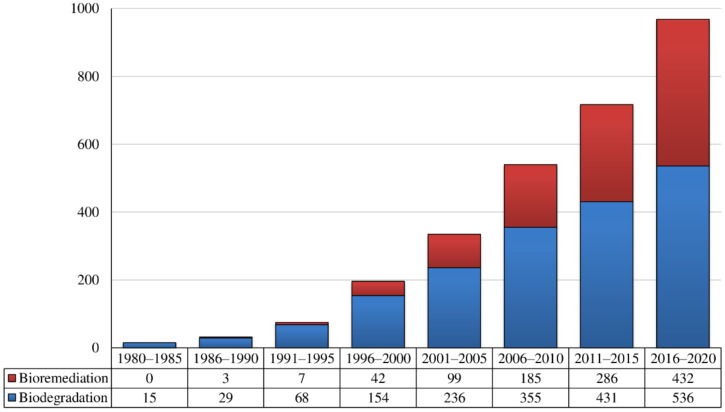
The number of articles published related to biodegradation and bioremediation involving rhodococci (available online at: http://www.scopus.com (accessed on 7 June 2021)). Queries: Title/Abstract/Keywords: *Rhodococcus*, Biodegradation, Bioremediation. Articles from the query results that were irrelevant were not counted.

**Figure 2 pathogens-10-00974-f002:**
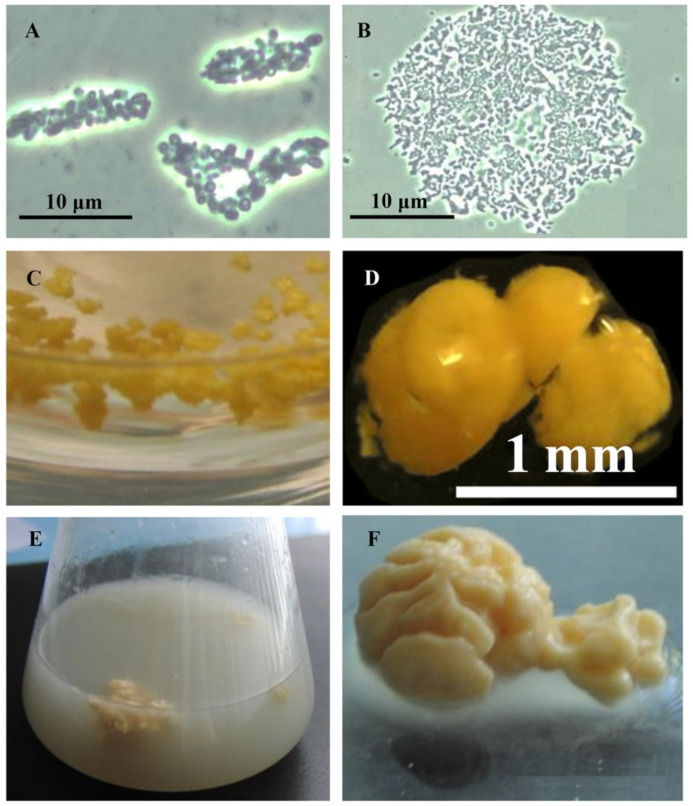
Cell aggregation of *R. ruber* IEGM 326 (**A**–**D**) and *R. erythropolis* IEGM 270 (**E**,**F**) grown in a mineral salt medium in the presence of 1 vol.% *n*-hexadecane. (**A**,**B**)—oblong (**A**) and rounded (**B**) cell microaggregates; (**C**,**D**)—microbial granules (macrocolonies); (**E**,**F**)—isolated cell aggregates.

**Figure 3 pathogens-10-00974-f003:**
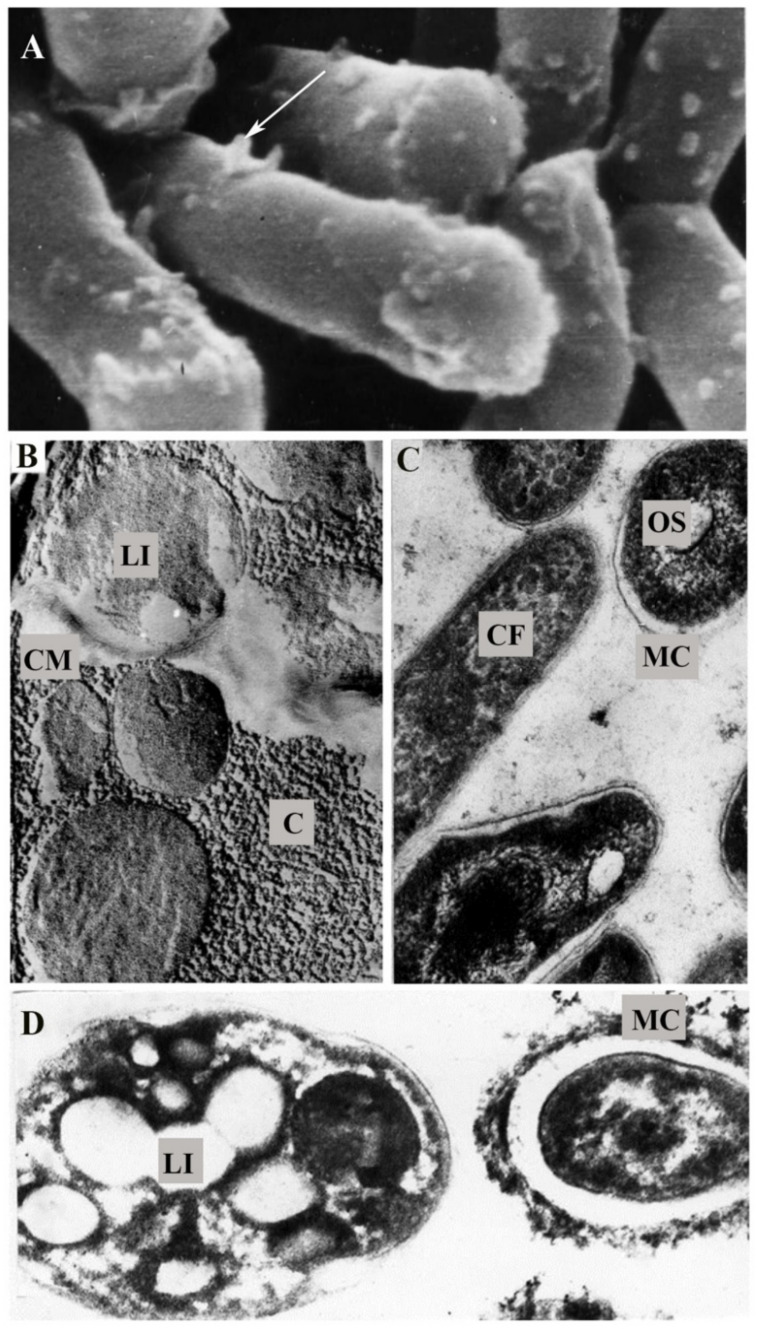
*R. ruber* cells grown in the presence of propane (**A**–**C**) and *n*-hexadecane (**D**). (**A**)—scanning electron microscope images of IEGM 333 cells, ×44,000, the arrow shows pineal appendages; (**B**)—cryofractography images of chips of IEGM 342 cells; (**C**)—ultrathin sections, strain IEGM 565, ×60,000; (**D**)—strain IEGM 333, ×85,000. LI—lipid inclusions; CM—cytoplasmic membrane; C—cytoplasm; OS—oxisome-like structures; FC—cytoplasm fragmentation; MC—microcapsule. In the presence of hydrocarbons, a *Rhodococcus* population contains both cells with and without microcapsules (**C**,**D**). A distinctive feature of cells is the fragmentation of the cytoplasm and the presence of numerous irregular semitransparent areas in it (**C**); a large number of lipid inclusions (**B**,**D**), the formation of oxisome-like structures, which are specialized sites for localization of oxidizing enzymes required for transformation of the hydrocarbon substrate (**C**). Modified from [23].

**Figure 4 pathogens-10-00974-f004:**
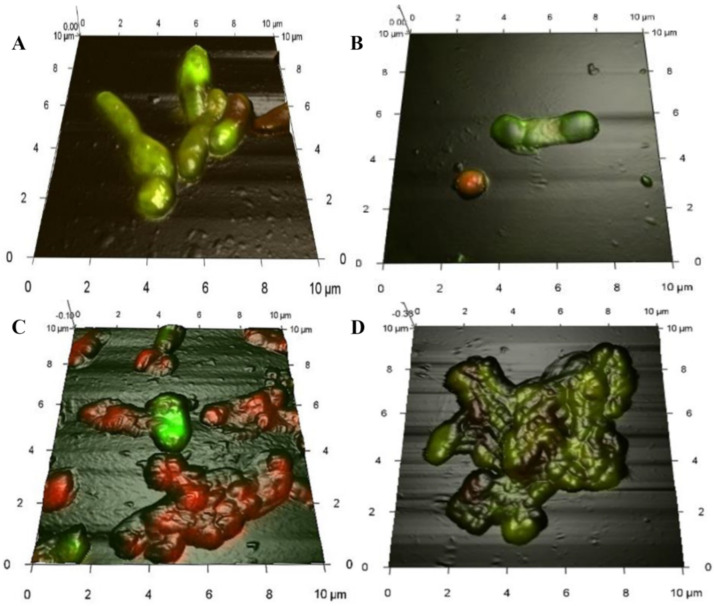
Combined 3D AFM/CLSM images of *R. ruber* IEGM 346 cells grown for 10 days in a mineral salt medium in the presence of glucose and diclofenac. (**A**)—medium with glucose; (**B**,**D**)—medium with glucose and 50 μg/L diclofenac; (**C**)—medium with glucose and 50 mg/L diclofenac. Damaged bacteria exhibit red fluorescence. Modified from [5].

**Figure 5 pathogens-10-00974-f005:**
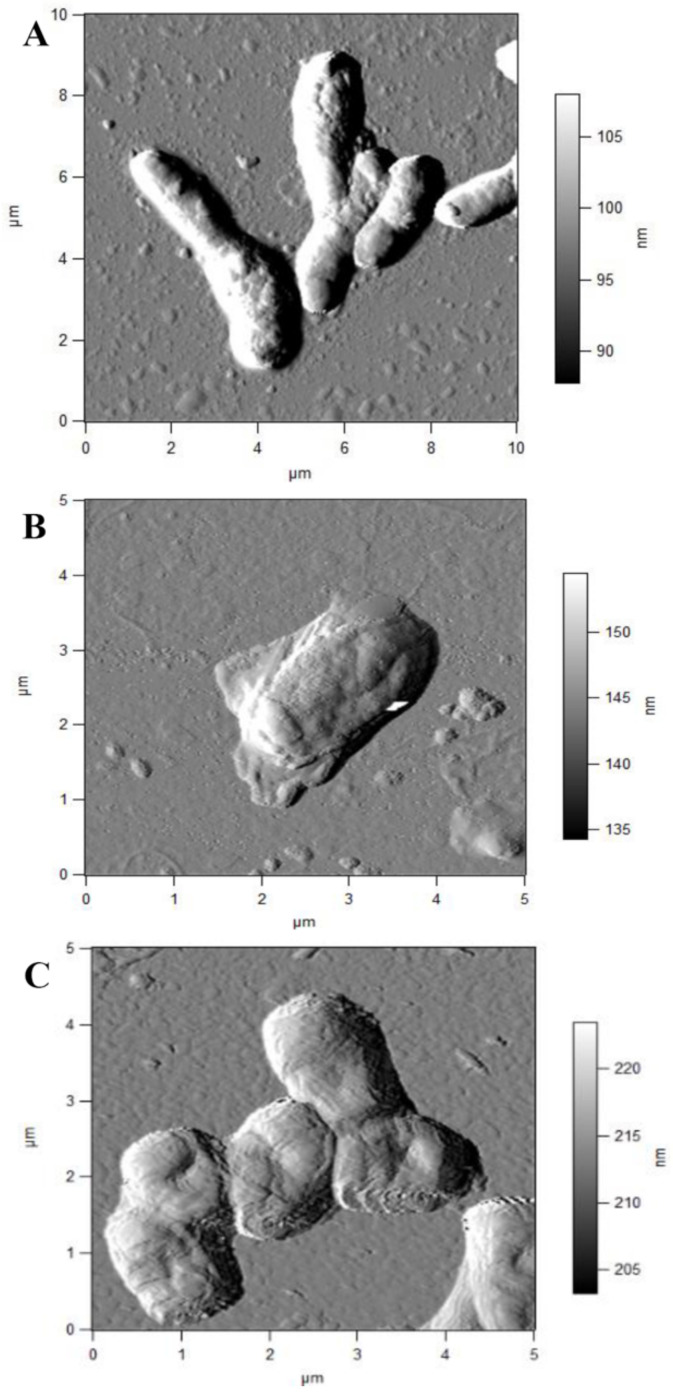
AFM images of *R. ruber* IEGM 346 cells grown for 10 days in a mineral salt medium supplemented with: (**A**)—glucose; (**B**)—glucose and 50 mg/L diclofenac; (**C**)—glucose and 50 μg/L diclofenac. Modified from [5].

**Table 1 pathogens-10-00974-t001:** Comparative morphometric parameters of rhodococci exposed to toxic compounds.

Strain	Variant	Length, μm	Width, μm	Volume, V, μm^3^	Area, S, μm^2^	S/V, μm^−1^	Roughness, nm
Diclofenac							
*R. ruber* IEGM 346	Control	3.0 ± 0.02	0.9 ± 0.05	1.9 ± 0.03	5.5 ± 0.05	2.9 ± 0.02	197.8 ± 2.30
	50 mg/L	3.5 ± 0,13	1.1 ± 0.02	3.3 ± 0.05	7.9 ± 0.10	2.4 ± 0.08	216.1 ± 5.51
	50 μg/L	2.2 ± 0.05	0.8 ± 0.01	1.0 ± 0.02	3.6 ±0.03	3.6 ± 0.02	249.6 ± 6.64
Betulin							
*R. rhodochrous* IEGM 66	Control	2.1 ± 0.30	0.8 ± 0.10	1.1 ± 0.39	6.3 ± 0.45	5.7 ± 0.69	268.5 ± 12.72
	500 mg/L	1.9 ± 0.47	0.7 ± 0.11	0.9 ± 0.33	5.4 ± 0.39	6.0 ± 0.85	359.6 ± 9.13
	3000 mg/L	1.8 ± 0.48	0.8 ± 0.12	0.9 ± 0.12	5.6 ± 1.07	6.2 ± 0.88	400.9 ± 7.92
Dehydroabietic acid							
*R. rhodochrous* IEGM 107	Control	1.3 ± 0.28	1.1 ± 0.19	1.3 ± 0.16	4.3 ± 0.28	3.2 ± 0.11	206.5 ± 10.72
	500 mg/L	1.8 ± 0.26	1.2 ± 0.24	2.2 ± 0.13	6.1 ± 0.23	2.7 ± 0.09	365.9 ± 6.92

**Table 2 pathogens-10-00974-t002:** Changes in zeta potential values of the cell surface of rhodococci exposed to complex organic substrates.

Strain	Variant	Zeta Potential
Dehydroabietic acid		
*R. rhodochrous* IEGM 107	Control	−26.6 ± 0.91
	500 mg/L	−27.3 ± 1.11
*R. erythropolis* IEGM 267	Control	−15.5 ± 1.42
	500 mg/L	−19.8 ± 1.23
Diclofenac		
*R. ruber* IEGM 346	Control	−35.3 ± 2.33
	50 mg/L	−31.3 ± 0.83
Betulin		
*R. rhodochrous* IEGM 66	Control	−34.8 ± 0.91
	3000 mg/L	−35.2 ± 1.11

## Data Availability

Data available within the article.

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
