# Peer review of "Responses to Ecopollutants and Pathogenization Risks of Saprotrophic Rhodococcus Species"

_pathogens, 2021, doi:10.3390/pathogens10080974_

Round 1
Reviewer 1 Report
The manuscript describes response of Rhodococcus species to presence of toxic compounds in the environment and possible appearance of their pathogenic features. I consider the topic very interesting. In my opinion, it deserves much better elaboration.
Major general comments:
- 1) My general comment is that the manuscript presents too many words with little effect.
- 2) Although it seems that pathogenization is the major topic in the manuscript, there is very little information concerning this process and the whole phenomenon is finally left in the mist. On several places it is mentioned that Rhodococcus may switch from saprophytic to pathogenic style of life but it is not explained what leads to this life strategy alteration. This key idea is not sufficiently supported by citations, explained and discussed.
- 3) Data concerning the switch of the life strategy of Rhodococcus from saprophytic to pathogenic lifestyle, the reasons for this and potential increasing risk due to polluted environment and use of Rhodococcus in biotechnology should be concentrated to a single coherent section. Now the mentions are scattered throughout the text.
- 4) There are many long, tangled, illogical and incomprehensible sentences
- 5) The authors jump in the text from one topic to another without order
- 6) There are many good ideas which, however, appear chaotically in the text
- 7) There are too many general statement with no lessons to the readers or are valid for all bacteria
- 8) The link between response to pollutants and development of pathogenic features is not explained
- 9) The contents of the sections frequently do not agree with the titles of the sections
- 10) The text should be radically shortened and should be focused on the topic presented in the title
- 11) There are many mistakes in English and incorrect expressions. I recommend the authors to use the professional English editing/proofreading service to check and correct the manuscript.
- 12) There are long sections without supporting the claims by citations
- 13) Many characteristics of Rhodococcus species are repeated in various parts of the text
- 14) There is more information on the topic in Abstract and Conclusion (lines 908-921) than within the main text. The conclusion should be much shorter, should not introduce new facts and ideas but summarize the main points explained in the text.
- 15) There might be a hypothesis at the end of the manuscript: The possible switch between saprophytic and pathogenic lifestyle is 1) a natural feature of Rhodococcus due to long evolution or 2) a new feature developed due to the evolution which is accelerated by the pollution of the environment. What is the role of biotechnology in this switch? I consider these questions the most interesting. It is also the reason why the topic deserves to be much better treated.
Specific comments:
The title should be improved to make it more clear and targeted
Lines 13-16: I guess, the key sentence in the manuscript, however its meaning is not clear
L 16-24: Long, complicated and incomprehensible sentence with more than 100 words.
L 26-29: Important idea which in not much clear or explained. "…Rhodococcus are among the first candidates, if further ecological changes, to move into the group.." This not clear to me.
L 50-51: representatives…evolution.. can be deleted
L 61: almost no competiors – many bacterial genera are suitable for used in biodegradation of pollutants: Pseudomonas, Achromobacter, Acinetobacter, Arthrobacter, Burkholderia, Geobacter, Mycobacterium
L 68: recalcitrant means: hard-to-degrade. Why to use it twice?
L 67-70: It is difficult to understand meaning of the sentence
L 70-74: A sentence with trivial information
L 91-11: No citation to support the claims
L 120: They… : who?
L 131-133: A sentence important for the main topic of the manuscript. But there is no citation.
L 146-160!: A single sentence? 150 words!
L 162-165: again no citation
L 176: aggression – against what?
L 183: in natural systems – may be deleted
L 185: mycolic acid-containing nocardioform actinomycetes – may be deleted
L 184-207: what is the relationship of this part to the main topic of the manuscript?
L 209-213: past tense should be used
L 224-225: It should be noted… of R. fascians. L 228-229: ..horizontal gene transfer…plant cells. A promising statement but no details are provided.
L 212-238: Why R. fascians is alternatively with and without quotation marks?
L 234- 235: Meaning of the sentence is not clear.
L 183, L 208, L 240: confusion in section titles: Two subsections concerning pathogenic species are within the section "Ubiquitous distribution of Rhodococcus in natural systems"
L 208-400: if pathogenicity is discussed, I would expect at the beginning of the subsections description which organisms are inflicted and what are the symptoms.
L 243: Why some species (names of the species?) are invalid? Should they be reclassified?
L 246: Please, explain how human introduced some species to environment
L 247: Why the habitats are optimal?
L 249-254: Description of the results of the detailed analysis of R. equi genome are confusing. I would expect the genome analysis may provide hints why some species may be pathogenic. Are there any clear differences in the presence of some genes between non-pathogenic and pathogenic species? In other words: Is it possible to differentiate such species according to the genome analysis? Is it possible to hypothesize how many genes are responsible for pathogenicity?
L 255-269: Many statements are not supported by citations
L 285: What makes R. equi a typical soil organism?
L 299-306: Finally, some mention concerning soil (saprophytic) Rhodococcus species which may convert to pathogenic. However, no details from references 72, 85, 87, 88 are provided. Why to avoid data which might support the main idea of the manuscript?
L 302-306: The risk of handling Rhodococcus spp. in biotechnology is not explained.
L 316-331: What is the connection of this part to pathogenicity?
L 332-335: Very unclear sentence
L 401: What are "biological features"? This should be maybe at the beginning of the text. Is the capability of Rhodococcus spp. to degrade various compounds considered "biological features" Which characteristic of these bacteria are not biological features?
L 402-429: citations are missing (106 is the only one)
L 614-621: The authors jump in text from specific to general topics without any order.
L 633-638: Again promising ideas connected to the main topic of the manuscript. But no connection and explanations are given.
L 924-934: Interesting points. Why these ideas are not properly explained and discussed in the text?
Author Response
Major general comments:
- 1) My general comment is that the manuscript presents too many words with little effect.
- 2) Although it seems that pathogenization is the major topic in the manuscript, there is very little information concerning this process and the whole phenomenon is finally left in the mist. On several places it is mentioned that Rhodococcus may switch from saprophytic to pathogenic style of life but it is not explained what leads to this life strategy alteration. This key idea is not sufficiently supported by citations, explained and discussed.
- 3) Data concerning the switch of the life strategy of Rhodococcusfrom saprophytic to pathogenic lifestyle, the reasons for this and potential increasing risk due to polluted environment and use of Rhodococcus in biotechnology should be concentrated to a single coherent section. Now the mentions are scattered throughout the text.
- 4) There are many long, tangled, illogical and incomprehensible sentences
- 5) The authors jump in the text from one topic to another without order
- 6) There are many good ideas which, however, appear chaotically in the text
- 7) There are too many general statement with no lessons to the readers or are valid for all bacteria
- 8) The link between response to pollutants and development of pathogenic features is not explained
- 9) The contents of the sections frequently do not agree with the titles of the sections
- 10) The text should be radically shortened and should be focused on the topic presented in the title
- 11) There are many mistakes in English and incorrect expressions. I recommend the authors to use the professional English editing/proofreading service to check and correct the manuscript.
- 12) There are long sections without supporting the claims by citations
- 13) Many characteristics of Rhodococcusspecies are repeated in various parts of the text
- 14) There is more information on the topic in Abstract and Conclusion (lines 908-921) than within the main text. The conclusion should be much shorter, should not introduce new facts and ideas but summarize the main points explained in the text.
- 15) There might be a hypothesis at the end of the manuscript: The possible switch between saprophytic and pathogenic lifestyle is 1) a natural feature of Rhodococcusdue to long evolution or 2) a new feature developed due to the evolution which is accelerated by the pollution of the environment. What is the role of biotechnology in this switch? I consider these questions the most interesting. It is also the reason why the topic deserves to be much better treated.
Answer. We express our great gratitude to the reviewer for a thorough analysis of our work. According to the fair critical comments, the text has been reduced by 29% (from 10,901 to 7,756 words, excluding references), sections not related to the topic of the review and repetitions are removed, the abstract and conclusion are changed, the style of the text is corrected. The corrected text focuses on the physiological properties of rhodococci (the ability of adhering and colonizing surfaces, complex life cycle, formation of resting cells and capsule-like structures, diauxotrophy and rigid cell wall), enhanced in the presence of hydrocarbon substrates that allow them to switch from saprophytic to pathogenic style of life. The necessary explanations and references have been added, as presented below in the responses to specific comments.
Specific comments:
The title should be improved to make it more clear and targeted
The title is modified.
Lines 13-16: I guess, the key sentence in the manuscript, however its meaning is not clear
We shortened and modify this sentence to clarify meaning.
L 16-24: Long, complicated and incomprehensible sentence with more than 100 words.
This sentence is also shortened and modified.
L 26-29: Important idea which in not much clear or explained. "…Rhodococcus are among the first candidates, if further ecological changes, to move into the group.." This not clear to me.
We modify the sentence to emphasize the role of increasing anthropogenic pressure on possible pathogenization of Rhodococcus.
L 50-51: representatives…evolution… can be deleted
Deleted
L 61: almost no competiors – many bacterial genera are suitable for used in biodegradation of pollutants: Pseudomonas, Achromobacter, Acinetobacter, Arthrobacter, Burkholderia, Geobacter, Mycobacterium
Changed to “not many competitors”.
L 68: recalcitrant means: hard-to-degrade. Why to use it twice?
“hard-to-degrade” is deleted
L 67-70: It is difficult to understand meaning of the sentence
The sentence is modified.
L 70-74: A sentence with trivial information
The sentence is deleted.
L 91-11: No citation to support the claims
The references are added.
L 120: They… : who?
Corrected.
L 131-133: A sentence important for the main topic of the manuscript. But there is no citation.
The reference is added.
L 146-160!: A single sentence? 150 words!
The sentence is shortened and modified.
L 162-165: again no citation
The reference is added.
L 176: aggression – against what?
The whole sentence is deleted.
L 183: in natural systems – may be deleted
Deleted.
L 185: mycolic acid-containing nocardioform actinomycetes – may be deleted
Deleted.
L 184-207: what is the relationship of this part to the main topic of the manuscript?
This short section briefly describes free-living Rhodococcus habitats in contrast to following sections on phytopathogenic and animal/human pathogens among rhodococci.
L 209-213: past tense should be used
Corrected.
L 224-225: It should be noted… of R. fascians. L 228-229: ..horizontal gene transfer…plant cells. A promising statement but no details are provided.
The details of these findings can be found in the references provided. The present study is focused on the free-living hydrocarbon-oxidizing rhodococci.
L 212-238: Why R. fascians is alternatively with and without quotation marks?
The taxonomy of R. fascians is still under development, but this is a valid species. We removed the quotation marks of R. fascians everywhere in the text.
L 234- 235: Meaning of the sentence is not clear.
The sentence is modified.
L 183, L 208, L 240: confusion in section titles: Two subsections concerning pathogenic species are within the section "Ubiquitous distribution of Rhodococcus in natural systems"
The section title is changed to “The ubiquity of Rhodococcus and the existence of pathogenic species”.
L 208-400: if pathogenicity is discussed, I would expect at the beginning of the subsections description which organisms are inflicted and what are the symptoms.
The subsection on phytopathogenic rhodococci begins with following information:
Rhodococci were detected in rhizosphere and phyllosphere microbiota [61−65]. Associated with plants, rhodococci can not only stimulate their productivity [66] but also cause pathology. Phytopathogens R. corynebacterioides and the R. fascians which are causative agents of bacteriosis in fruit and berry plants were described. Initially, R. fascians was an epiphyte that turned into an endophyte [67,68]. R. fascians induces fasciation and leafy gall disease in a variety of plants, viz. 87 genera spanning 40 plant families, including the newly emerging Pistachio Bushy Top Syndrome (PBTS) [67,69−74].
The subsection on pathogenic rhodococci is modified and begins with following information:
Typical soil rhodococci “R. equi” have long been isolated from the lung tissues of 3-month-old foals and are now known as a facultative intracellular pathogen not only in young domestic animals but also as a serious opportunistic human pathogen. For many years, “R. equi” representatives were isolated from soils of all continents, except for Antarctica. They are the causative agents of slowly progressing pneumonia (”rhodococcal pneumonia”) and subcutaneous abscesses most frequently observed in HIV-infected patients with a compromised immune system [77−80]. Several cases of nosocomial infections caused by “R. equi” (meningitis, pneumonia, endocarditis, keratitis) have been described [33,81,82].
L 243: Why some species (names of the species?) are invalid? Should they be reclassified?
Species with a validly published name are included into the Approved List of Bacterial Names, while invalid species, not validly published or not meeting the minimal requirements of the Prokaryotic Code, are given in quotation marks.
L 246: Please, explain how human introduced some species to environment
The whole paragraph is removed.
L 247: Why the habitats are optimal?
The above paragraph is removed.
L 249-254: Description of the results of the detailed analysis of R. equi genome are confusing. I would expect the genome analysis may provide hints why some species may be pathogenic. Are there any clear differences in the presence of some genes between non-pathogenic and pathogenic species? In other words: Is it possible to differentiate such species according to the genome analysis? Is it possible to hypothesize how many genes are responsible for pathogenicity?
Based on the performed analysis of R. equi genomes published, hallmarks that distinguish this pathogenic species from non-pathogenic rhodococci were identified. R. equi is genetically homogeneous, it has a small (about 5.0 Mb) genome, which lacks the extensive catabolic and secondary metabolic components (for example, phosphoenolpyruvate: carbohydrate transport system) of non-pathogenic environmental rhodococci. The pathogenicity of R. equi is mediated by Virulence Associated Proteins (VAPs) encoded on acquired pathogenicity islands (PAIs) of host-adapted conjugative virulence plasmids. It is the presence of VAP plasmids that determines this species as pathogenic.
The phylogenetic analysis proved that Rhodococcus spp. (both pathogenic and non-pathogenic species) have stable core genomes. There are genes involved in protein biosynthesis, metabolism, transport, and signal transduction which can be associated with both adaptation and virulence. It is known that virulence-related genes in R. equi are conserved in environmental rhodococci or have homologs in nonpathogenic actinomycetes. Since R. equi have a high degree of core genome similarity with non-pathogenic environmental Rhodococcus species, R. equi presumably evolved due to the cooption of core actinobacterial functions, originally selected in a non-host environment. The acquisition of critical components (for example, VAP plasmid) through horizontal gene transfer leads to the transition of a commensal organism into a pathogen due to the cooptation of preexisting virulence-associated core genes.
The information on R. equi genome was added to the updated subsection 2.3. Animal and human Rhodococcus pathogens.
L 255-269: Many statements are not supported by citations
The references 25−27; 14,20,26,28−30; 33−38; 22,35 are placed after corresponding statements in the text.
L 285: What makes R. equi a typical soil organism?
R. equi is a normal inhabitant of soil and “R. equi” representatives were isolated from soils of all continents, except for Antarctica.The overall features of the organism appear to be versatility and capacity to adapt to environments as diverse as soil and the mammalian body (ref. 86 Vázquez-Boland, J.A.; Prescott, J.F.; Meijer, W.G.; Leadon, D.P.; Hines, S.A. Havermeyer workshop report: Rhodococcus equi comes of age. Equine Vet. J. 2009, 41, 93−95). In addition to its pathogenic life-style, R. equialso thrives as a soil-dwelling microorganism capable of rapid growth in soil and manure using steroids, such as cholesterol, as sole carbon and energy sources (van der Geize R, Grommen AWF, Hessels GI, Jacobs AAC, Dijkhuizen L (2011) The Steroid Catabolic Pathway of the Intracellular Pathogen Rhodococcus equi Is Important for Pathogenesis and a Target for Vaccine Development. PLoS Pathog 7(8): e1002181. https://doi.org/10.1371/journal.ppat.1002181).
L 299-306: Finally, some mention concerning soil (saprophytic) Rhodococcus species which may convert to pathogenic. However, no details from references 72, 85, 87, 88 are provided. Why to avoid data which might support the main idea of the manuscript?
These articles contain only descriptions of diseases caused by pathogenic bacteria and methods of their treatment, while there is no data that could support the main idea of the manuscript. Here, the main idea may be that there are really many pathogenic species, they simply have not been identified before due to the difficulties of identifying this group of actinomycetes and the lack of appropriate methods in routine clinical laboratories [63(former 72),33(former 85),34(former 87),98(former 88)].
This comment is added in the text.
L 302-306: The risk of handling Rhodococcus spp. in biotechnology is not explained.
We have described the risk of pathogenization of environmentally relevant Rhodococcus spp., which can be used in ecobiotechnologies for the biodegradation of recalcitrant pollutants. Since this review does not provide any information about specific biotechnological applications of rhodococci, we cannot analyze such risks in depth. However, it may be a good idea for further studies.
L 316-331: What is the connection of this part to pathogenicity?
This part was removed.
L 332-335: Very unclear sentence
This sentence was removed together with the whole part.
L 401: What are "biological features"? This should be maybe at the beginning of the text. Is the capability of Rhodococcus spp. to degrade various compounds considered "biological features" Which characteristic of these bacteria are not biological features?
This part was removed according to the reviewer’s comments.
L 402-429: citations are missing (106 is the only one)
This part was removed.
L 614-621: The authors jump in text from specific to general topics without any order.
We modified this text to better explain that neutral lipid storage, as a source of endogenous water, allows bacteria to tolerate soil drought and high temperatures. This statement is supported by references 7,30, 8,12,52,121.
Rhodococcus are capable of shifting to specific resting (viable but nonculturable, VBNC) forms [11,120], and accumulate large amounts of storage substances, for example, triacylglycerols, esters wax, and polyhydroxyalkanoate, as reservoirs of carbon and chemical energy [52,103,121]. The accumulated neutral lipids are not only a basis for the biosynthesis of mycolic acids and regulation of membrane fluidity but also a source of endogenous water [7,30], which allows rhodococci to tolerate soil drought and high temperatures [8,12,52,121].
L 633-638: Again promising ideas connected to the main topic of the manuscript. But no connection and explanations are given.
The possibility of MDR-coding plasmid pRErm46 transferring from R. equi to other actinobacterial species is emphasized with the relevant reference 89. Since the quorum-quenching activity of Rhodococcus is not directly connected with its possible pathogenicity, this property is only mentioned as an example of degrading activity and competitiveness, and provided with references 124,125.
L 924-934: Interesting points. Why these ideas are not properly explained and discussed in the text?
Yes, it is interesting idea. Indeed, we attempted to show that adaptation features of free-living saprotrophic Rhodococcus, such as the ability of adhering and colonizing surfaces, complex life cycle, formation of resting cells and capsule-like structures, diauxotrophy and rigid cell wall, developed against the negative effects of anthropogenic pollutants, can be considered as “universal” pathogenic factors. Therefore, under the increasing environmental pollution, rhodococci are among the potentially dangerous microorganisms with “unprofessional” parasitism.
Reviewer 2 Report
Ivshina et al's review paper "Response Mechanisms to Ecopollutants and Pathogenization Risks of Saprotrophic Rhodococcus" is a comprehensive information collection. My concerns are:
(1) The title is not appropriate. Only one section in the paper is talking about the mechanism which is section IV "Adaptive сell modifications of rhodococci exposed to hydrocarbons and other environmental pollutants". 75% of the content is about the nature of rhodococci which has no business with "mechanisms".
(2) The English of the manuscript needs to be upgraded. Although, I do not observe big grammar mistakes or typos. When I am reading, it just sounds non-native. I strongly suggest the whole essay be revised by a native English speaker.
(3) I do not fully agree with the content of Line 82-85. Most heterotrophic bacteria can degrade organic molecules. I do not think it is the major reason that rhodococci can tolerate harsh conditions.
(4) For Line 107-110, I do not see a logical flow here. Why should you mention this information here?
(5) Line 114, I am lost here. How can novel strains be designed?
Author Response
We thank the reviewer for a thorough analysis of the manuscript. In accordance with the comments, the following corrections were made to the text.
Comments and Suggestions for Authors
Ivshina et al's review paper "Response Mechanisms to Ecopollutants and Pathogenization Risks of Saprotrophic Rhodococcus" is a comprehensive information collection. My concerns are:
(1) The title is not appropriate. Only one section in the paper is talking about the mechanism which is section IV "Adaptive сell modifications of rhodococci exposed to hydrocarbons and other environmental pollutants". 75% of the content is about the nature of rhodococci which has no business with "mechanisms".
Answer: (1) The title is changed.
(2) The English of the manuscript needs to be upgraded. Although, I do not observe big grammar mistakes or typos. When I am reading, it just sounds non-native. I strongly suggest the whole essay be revised by a native English speaker.
Answer: The text is shortened and revised by a native English speaker.
(3) I do not fully agree with the content of Line 82-85. Most heterotrophic bacteria can degrade organic molecules. I do not think it is the major reason that rhodococci can tolerate harsh conditions.
Answer: Yes, most heterotrophic bacteria can degrade organic molecules. However, not many bacteria can degrade hydrocarbons and their derivatives, considered as recalcitrant compounds. We think that due to the hydrocarbon-oxidizing ability, rhodococci formed such specific cellular features as rigid lipophilic cell wall and adherence to hydrophobic substances, which help them to tolerate harsh conditions, similarly to other hydrocarbon-oxidizing bacteria Pseudomonas, Arthrobacter, Burkholderia, Mycobacterium etc.
(4) For Line 107-110, I do not see a logical flow here. Why should you mention this information here?
Answer: Since the studies reported in this section (and generally in the review) are performed with Rhodococcus strains from the IEGM collection, we provided the information on this collection. The sentence is shortened and modified.
(5) Line 114, I am lost here. How can novel strains be designed?
Answer: This part is modified and the phrase on novel strains to be designed is removed.
Round 2
Reviewer 1 Report
The authors did a good job, responded to the most of the comments, modified the manuscript and added necessary references. The manuscript was improved considerably. I appreciate rigorous highlighting of the corrected part.
Reviewer 2 Report
I have no further questions.